# Keratoconjunctivitis Sicca in Sjögren Disease: Diagnostic Challenges and Therapeutic Advances

**DOI:** 10.3390/ijms26188824

**Published:** 2025-09-10

**Authors:** Muhammad Soyfoo, Elie Motulsky, Julie Sarrand

**Affiliations:** 1Department of Rheumatology, Hôpital Universitaire de Bruxelles (HUB), 1070 Bruxelles, Belgium; julie.sarrand@ulb.be; 2Department of Ophthalmology, Hôpital Universitaire de Bruxelles (HUB), 1070 Bruxelles, Belgium; elie.motulsky@hubruxelles.be

**Keywords:** Sjögren’s disease, sicca, keratoconjunctivitis, interferon, cytokines

## Abstract

Keratoconjunctivitis sicca (KCS), also commonly known as dry eye disease (DED), is one of the most prevalent and crippling features of Sjögren disease (SD), a chronic systemic autoimmune disorder featuring lymphocytic infiltration and progressive impairment of exocrine glands. KCS affects up to 95% of patients with SD and is often the earliest and most persistent manifestation, significantly compromising visual function, ocular comfort, and overall quality of life. Beyond the ocular surface, KCS mirrors a wider spectrum of immune dysregulation and epithelial damage characteristic of the disease, making it a valuable window into the underlying systemic pathology. The pathophysiology of KCS in SD is complex and multifactorial, involving an interplay between autoimmune-mediated lacrimal gland dysfunction, neuroimmune interactions, ocular surface inflammation, and epithelial instability. Tear film instability and epithelial injury result from the aberrant activation of innate and adaptive immunity, involving T and B lymphocytes, pro-inflammatory cytokines, and type I interferon pathways. Despite the clinical significance of KCS, its diagnosis remains challenging, with frequent discrepancies between subjective symptoms and objective findings. Traditional diagnostic tools often lack sensitivity and specificity, prompting the development of novel imaging techniques, tear film biomarkers, and standardized scoring systems. Concurrently, therapeutic strategies have evolved from palliative approaches to immunomodulatory and regenerative treatments, aiming to restore immune homeostasis and epithelial integrity. This review provides a comprehensive update on the pathogenesis, diagnostic landscape, and emerging treatments of KCS in the context of SD.

## 1. Introduction

Sjögren disease (SD) is a chronic systemic autoimmune disorder characterized by lymphoplasmacytic infiltration of exocrine glands, leading to the hallmark sicca symptoms of dry eyes and dry mouth [1,2]. Although its pathophysiology remains incompletely understood, it is thought to involve abnormal lymphocyte activation, pro-inflammatory cytokine secretion, and type I interferon pathway activation [3,4]. With an estimated prevalence of 0.5–1% and a striking female predominance of 9:1, SD ranks among the most common systemic autoimmune diseases [5,6].

Exocrine gland dysfunction is the clinical hallmark of SD, with keratoconjunctivitis sicca (KCS) representing one of its most frequent and debilitating manifestations. Ocular involvement occurs in up to 95% of patients and often constitutes the presenting symptom leading to diagnosis [7,8,9,10]. KCS is a multifactorial process in which autoimmune inflammation, altered tear film composition, and impaired ocular surface integrity interact to create a self-perpetuating cycle of dysfunction. Beyond the individual burden, KCS in SD imposes a substantial socioeconomic impact, with annual healthcare costs exceeding 5.9 billion dollars in the United States and quality-of-life reductions comparable to those observed in chronic pain conditions [11,12].

The pathogenesis of SD reflects a complex interplay of genetic and environmental factors. Associations with HLA alleles, particularly HLA-DR3 and HLA-DQ2, support a strong genetic contribution, while viral infections such as Epstein–Barr virus may act as environmental triggers in genetically susceptible individuals [13,14,15].

Recent advances in ocular surface immunology, tear film biology, and inflammatory mechanisms have reshaped the understanding of KCS in SD. Novel imaging tools now permit in-depth evaluation of the meibomian gland and ocular surface structure, while emerging targeted therapies modulate specific inflammatory pathways. This review summarizes current knowledge of ocular manifestations in SD, highlights diagnostic innovations, and discusses evolving therapeutic strategies, with the goal of providing clinicians and researchers with an updated perspective on the best practices and future directions in KCS management.

## 2. Pathophysiology of Keratoconjunctivitis Sicca in Sjögren Disease

### 2.1. Autoimmune Mechanisms and Glandular Dysfunction

The pathophysiology of keratoconjunctivitis sicca (KCS) in Sjögren disease extends beyond simple tear deficiency and reflects a complex autoimmune cascade [16,17,18]. The process is initiated by infiltration of activated T lymphocytes and other immune cells into the lacrimal glands, producing focal lymphocytic sialadenitis. These infiltrates, composed mainly of CD4^+^ T-helper cells, B cells, and plasma cells, progressively disrupt acinar and ductal architecture, impairing tear secretion and driving chronic glandular dysfunction [19,20].

The initiating factors in Sjögren disease remain unclear, though molecular mimicry between viral and glandular self-antigens has been implicated [21]. This loss of tolerance favors autoimmune responses against exocrine tissues, with Ro52 and Ro60—major constituents of the Ro/SSA complex—emerging as dominant targets expressed in lacrimal glands [22,23].

The inflammatory process in the lacrimal glands follows a characteristic pattern of progression. Early stages are characterized by periductal lymphocytic infiltration, with preservation of acinar architecture. As the disease progresses, the inflammatory infiltrate expands, forming organized lymphoid structures that may resemble germinal centers [24]. This tertiary lymphoid tissue formation is associated with local production of autoantibodies and pro-inflammatory cytokines [25].

The autoimmune process is characterized by the production of specific autoantibodies, including anti-Ro/SSA and anti-La/Sjögren Syndrome Antigen B (SSB) antibodies, which are present in approximately 60–70% of SD patients [24,26,27]. These antibodies not only serve as diagnostic markers but also contribute directly to tissue damage through complement activation and immune complex formation. Anti-Ro52 antibodies may be particularly important in lacrimal gland dysfunction, as they can interfere with cellular calcium homeostasis and induce apoptosis in acinar cells [28].

In SD, lacrimal gland injury arises from Fas–FasL–induced apoptosis, cytotoxic T-cell activity, and cytokine-mediated inflammation. TNF-α, IL-1β, and IFN-γ sustain a pro-inflammatory niche that perpetuates destruction and hinders repair [29,30].

Type I interferons are central to the pathogenesis of Sjögren disease (SD), with many patients exhibiting an “interferon signature” in their gene expression profiles [31,32]. This activation may be triggered by viral infections or endogenous nucleic acids released from damaged cells. Interferon signaling promotes immune activation, autoantibody production, and tissue injury, thereby representing an attractive therapeutic target [33].

The progression of lacrimal gland dysfunction in SD appears to follow a temporal sequence, in which molecular and cellular alterations precede overt clinical manifestations [34]. Single-cell transcriptomic analyses of murine lacrimal glands have revealed complex changes in epithelial cell populations, including distinct transcriptional alterations in acinar and ductal cells that occur before gross morphological damage becomes evident [35]. Early non-immunologic insults to the glandular microenvironment—such as basement membrane disruption, myoepithelial cell dysfunction, and altered neural innervation—may create a permissive niche for subsequent autoimmune attack [36,37].

Neuroendocrine and hormonal influences further shape disease susceptibility. Dysregulation of the hypothalamic–pituitary–adrenal (HPA) axis, including reduced basal adrenocorticotropic hormone (ACTH) levels and altered cortisol responses, impairs glandular homeostasis in SD patients [38,39]. Androgen deficiency, particularly decreased dehydroepiandrosterone (DHEA) and its sulfate conjugate (DHEA-S), has been linked to reduced structural integrity and secretory activity of lacrimal glands. These hormonal alterations contribute to both glandular dysfunction and immune dysregulation, potentially explaining the strong female predominance and increased susceptibility after menopause [40,41,42,43].

At the cellular level, lacrimal gland epithelial cells undergo apoptosis through both intrinsic mitochondrial and extrinsic death receptor pathways, with activation of caspases-3, -8, and -9 leading to progressive acinar cell loss [44,45]. Chronic activation of the unfolded protein response (UPR) in the endoplasmic reticulum, particularly under the influence of IFN-γ, results in sustained cellular stress, misfolded protein accumulation, and eventual cell death [3,46,47,48]. This failure of adaptive stress responses exacerbates secretory dysfunction and perpetuates ocular surface damage.

### 2.2. Tear Film Abnormalities and Ocular Surface Changes

The tear film in Sjögren disease (SD) patients shows profound qualitative and quantitative abnormalities that extend beyond simple aqueous deficiency [4,49]. The normal tear film comprises three layers: an outer lipid layer secreted by the meibomian glands, a central aqueous layer produced by the lacrimal glands, and an inner mucin layer derived from conjunctival goblet cells. In SD, dysfunction affects all three layers, creating a complex and multifactorial disturbance of tear film homeostasis [50].

The aqueous layer deficiency, while the most clinically apparent, is accompanied by significant compositional changes. Tear protein concentration increases due to reduced volume, while key protective components such as lactoferrin, lysozyme, and secretory IgA are decreased [51,52]. These alterations compromise antimicrobial defense and disturb osmotic balance at the ocular surface.

Meibomian gland dysfunction (MGD) is also common in SD and contributes to evaporative dry eye that compounds aqueous deficiency [53,54]. Similarly to the lacrimal glands, meibomian glands are affected by inflammatory processes, leading to altered lipid composition and reduced lipid layer stability. The result is increased tear evaporation and further ocular surface desiccation [55].

Finally, mucin layer abnormalities arise from goblet cell loss and altered mucin production [56]. Goblet cell density is markedly reduced in SD, and surviving cells may secrete mucins with altered biochemical properties [57]. These changes diminish tear wettability and weaken the protective barrier function of the ocular surface.

### 2.3. Inflammatory Cascade and Ocular Surface Damage

The ocular surface in SD becomes a site of chronic inflammation that perpetuates and amplifies the initial autoimmune process [58,59]. Hyperosmolarity resulting from tear deficiency activates inflammatory pathways in the corneal and conjunctival epithelium, leading to the production of inflammatory mediators including matrix metalloproteinases (MMPs), cytokines, and chemokines [60]. Tear film hyperosmolarity (>300 mOsm/L) initiates a complex cascade of inflammatory events that begins with immediate cellular responses to osmotic stress and progresses to sustained inflammatory activation [61,62]. When tear osmolarity exceeds that of epithelial cells, water efflux occurs, causing cell shrinkage and triggering multiple stress-responsive signaling pathways [63,64]. Hyperosmotic stress directly activates the NLRP3 inflammasome complex in corneal and conjunctival epithelial cells, leading to caspase-1 activation and subsequent proteolytic processing of pro-IL-1β and pro-IL-18 into their mature, biologically active forms [65,66]. This process is accompanied by enhanced reactive oxygen species (ROS) generation from damaged mitochondria, which further amplifies inflammasome activation and creates a self-perpetuating cycle of inflammatory mediator release [67,68]. Osmotic stress triggers rapid nuclear translocation of NF-κB, with the degree of translocation directly proportional to tear osmolarity levels [69,70]. Activated NF-κB upregulates transcription of multiple inflammatory genes, including TNF-α, IL-1β, IL-6, and matrix metalloproteinases (MMP-3 and MMP-9), which collectively contribute to epithelial barrier dysfunction and tissue damage [71,72]. The NF-κB pathway also induces expression of adhesion molecules such as ICAM-1 and selectins, facilitating immune cell recruitment to the ocular surface [59,73]. Recent discoveries have highlighted the role of the cGAS-STING pathway in hyperosmolarity-induced inflammation, where oxidized mitochondrial DNA released from stressed cells activates cytosolic DNA sensors, leading to type I interferon production and amplification of inflammatory responses [74,75]. This pathway represents a critical link between cellular stress and innate immune activation in dry eye disease [76].

Hyperosmolarity induces specific changes in epithelial cell function, including disruption of tight junction proteins (particularly occludin and claudin-1), loss of barrier function, and altered mucin production patterns [77,78]. Goblet cells are particularly susceptible to osmotic stress, undergoing apoptosis through ER stress-mediated pathways and showing reduced mucin synthesis even before cell death occurs, contributing to tear film instability [15,79]. The combination of epithelial damage, inflammatory mediator release, and compromised barrier function creates a pathological feedback loop that perpetuates and amplifies the dry eye condition [4,80].

The complement system plays a crucial role in ocular surface damage in SD. Complement activation occurs through both classical and alternative pathways, leading to the formation of membrane attack complexes that directly damage epithelial cells [62,63]. This process is particularly pronounced in areas of epithelial defects, where complement deposition can be demonstrated histologically.

Epithelial cell death occurs through multiple mechanisms, including apoptosis, necrosis, and autophagy [71]. The loss of epithelial cells compromises barrier function, allowing for increased penetration of inflammatory mediators and potential pathogens. This creates a vicious cycle where epithelial damage promotes inflammation, which in turn leads to further epithelial loss [81].

The corneal nerve damage observed in SD patients contributes to both sensory abnormalities and impaired reflex tearing [82,83]. Inflammatory mediators can directly damage corneal nerve fibers, leading to reduced corneal sensitivity and altered blink patterns. This neurogenic component of KCS adds another layer of complexity to the pathophysiology and may explain why some patients experience severe symptoms despite apparently adequate tear production [84] (Figure 1).

## 3. Diagnostic Challenges in Keratoconjunctivitis Sicca

### 3.1. Clinical Presentation and Symptom Variability

The diagnosis of KCS in SD presents numerous challenges, beginning with the highly variable and often nonspecific nature of presenting symptoms [7,80]. Patients may report a wide range of ocular complaints, including burning, stinging, foreign body sensation, photophobia, blurred vision, and paradoxical tearing [85,86]. The severity of symptoms does not always correlate with objective findings, making clinical assessment particularly challenging [87,88].

The discordance between symptoms and signs in SD patients is particularly pronounced and represents a unique diagnostic challenge [89,90]. Patients may experience severe discomfort with minimal objective findings, while others may have significant corneal staining with relatively mild symptoms [91]. This phenomenon may be related to altered corneal sensitivity, which is common in SD and can affect both symptom perception and protective reflexes [92,93].

The temporal pattern of symptoms can provide diagnostic clues, with many SD patients experiencing worsening symptoms throughout the day as tear production decreases and environmental factors accumulate [94,95]. Morning symptoms may be relatively mild, while evening discomfort can be severe. However, some patients may experience the opposite pattern, particularly if concurrent MGD is present, as meibomian gland dysfunction may be worse in the morning after prolonged eyelid closure [96].

Symptom triggers are highly individualized but commonly include environmental factors such as air conditioning, wind, low humidity, and visual tasks requiring sustained attention [97,98]. Digital device use has become an increasingly important trigger, as reduced blink rates during screen time can exacerbate tear film instability [99,100]. Understanding these triggers is crucial for both diagnosis and management, as symptom patterns may vary seasonally or with occupational exposures [86].

The psychological impact of chronic KCS cannot be underestimated and adds another layer of complexity to the diagnostic process [12,101]. Many patients experience anxiety, depression, and social isolation related to their ocular symptoms [102,103]. The unpredictable nature of symptom fluctuations can lead to anticipatory anxiety, while the chronic nature of the condition may result in depression and social withdrawal [104]. These psychological factors can influence symptom reporting and may complicate the diagnostic process, particularly when objective findings appear minimal compared to subjective complaints [105].

Sleep disturbances are common in SD patients and may be both a cause and consequence of ocular symptoms [106,107]. Nocturnal lagophthalmos, reduced tear production during sleep, and morning symptoms can create a cycle of poor sleep quality and worsening daytime symptoms. The relationship between sleep and ocular surface health is bidirectional, as poor sleep quality can exacerbate inflammatory processes and reduce pain tolerance [108,109].

### 3.2. Diagnostic Testing Limitations and Interpretation

Traditional diagnostic tests for KCS, while valuable, have significant limitations when applied to SD patients. The Schirmer test, which measures tear production over a defined time period, can be influenced by reflex tearing, ambient conditions, and patient anxiety [110]. Normal values may be observed in some SD patients, particularly in early disease stages, while abnormal results may occur in patients without systemic autoimmune disease.

Tear break-up time (TBUT) testing provides information about tear film stability but can be affected by factors beyond tear composition, including eyelid abnormalities, surface irregularities, and patient cooperation [111]. The test requires skilled interpretation and may show significant inter-observer variability.

Corneal and conjunctival staining with vital dyes such as fluorescein and lissamine green reveals epithelial damage patterns but may not correlate well with symptom severity [10]. The interpretation of staining patterns requires experience, and mild staining may be overlooked while severe staining may not reflect the full extent of ocular surface compromise.

Tear osmolarity testing has emerged as a valuable diagnostic tool, as hyperosmolarity is a key feature of KCS [64]. However, osmolarity values can fluctuate throughout the day and may be influenced by factors such as recent tear instillation, environmental conditions, and systemic hydration status. Single-point measurements may not capture the full extent of osmotic instability.

### 3.3. Differential Diagnosis Considerations

The diagnosis of KCS in SD must consider numerous other conditions that can present with similar symptoms and signs. Allergic conjunctivitis, infectious conjunctivitis, and medicamentosa from chronic topical medication use can all mimic aspects of KCS. The presence of concurrent conditions further complicates diagnosis, as many SD patients may have multiple ocular surface disorders simultaneously.

Age-related changes in tear production and composition can overlap with early SD manifestations, making diagnosis particularly challenging in older patients. Hormonal influences, particularly in postmenopausal women, can affect tear production and may confound the assessment of autoimmune-related changes [26].

Medication-induced dry eye from systemic drugs such as antihistamines, antidepressants, and diuretics must be considered in the differential diagnosis. Many SD patients are prescribed medications that can exacerbate ocular surface symptoms, creating a complex interplay between disease-related and medication-related factors.

Contact lens wear can significantly alter ocular surface characteristics and may mask or exacerbate underlying KCS. The assessment of contact lens-wearing SD patients requires careful consideration of lens-related factors and may necessitate a period of lens discontinuation for accurate diagnosis [112].

## 4. Advanced Diagnostic Approaches

A summary of the diagnostic modalities used in KCS, along with their targets, advantages, and limitations, is presented in Table 1.

### 4.1. Imaging Technologies and Biomarkers

Recent advances in ocular imaging have revolutionized the diagnostic approach to KCS in SD. Meibography, which provides detailed visualization of meibomian gland structure, has revealed the high prevalence of MGD in SD patients [113]. This non-invasive technique allows for quantitative assessment of gland dropout and morphological changes, providing objective evidence of lipid layer dysfunction. Advanced meibography systems now provide automated calculation of dropout scores using planimetry software to measure the percentage of gland area loss relative to total tarsal area [114,115]. The dropout score correlates directly with functional measures of meibomian gland dysfunction, with scores >30% associated with significant tear film instability and scores >60% indicating severe functional impairment [113,116]. Importantly, upper and lower eyelid dropout patterns differ significantly, with upper lids showing more frequent morphological abnormalities while lower lids demonstrate greater absolute dropout areas [53,117].

Tortuosity index (TI) represents a quantitative measure of meibomian gland shape deviation from normal linear morphology, calculated as the ratio of actual gland path length to the straight-line distance between gland endpoints [118,119]. Normal healthy individuals exhibit TI values < 0.1, while patients with meibomian gland dysfunction show significantly elevated TI values, with each 0.1 increase associated with reduced tear break-up time and increased dry eye symptoms [118,119].

Gland tortuosity often represents the earliest morphological change in meibomian gland dysfunction, preceding visible dropout by months to years, making it a valuable early diagnostic marker [120,121]. The mechanism underlying increased tortuosity involves inflammatory changes in the tarsal plate that alter the mechanical environment of the glands, forcing them to follow tortuous paths as they attempt to maintain their length within a contracting fibrous matrix [122,123]. High tortuosity indices (>0.2) significantly impair meibum expression efficiency during blinking, as the increased path length and multiple directional changes create resistance to flow [124,125].

Modern automated analysis systems provide multiple additional parameters including diameter deformation index (DI), which measures width variations along gland length, and signal index (SI), which reflects gland density and structural integrity based on infrared reflectance patterns [126,127]. These parameters show excellent inter-observer reliability (ICC > 0.8) and provide complementary information to traditional dropout scores, with combined assessment significantly improving diagnostic accuracy for meibomian gland dysfunction [128,129].

Longitudinal studies demonstrate that meibographic parameters predict treatment response, with patients showing high initial tortuosity indices and extensive dropout being less likely to respond to conventional therapies and more likely to require advanced interventions such as intense pulsed light or thermal pulsation treatments [130,131]. The combination of upper eyelid tortuosity and lower eyelid dropout provides the strongest correlation with patient-reported symptoms and objective clinical measures, supporting the importance of comprehensive bilateral assessment in clinical practice [132,133].

Infrared meibography is particularly valuable as it can visualize gland morphology through the eyelid without requiring eyelid eversion. This technique has revealed that meibomian gland dropout in SD patients follows characteristic patterns, with more severe involvement typically seen in the central portions of the eyelids. The correlation between meibography findings and clinical measures of MGD provides valuable insights into the pathophysiology of evaporative dry eye in SD.

In vivo confocal microscopy (IVCM) provides high-resolution visualization of corneal and conjunctival surfaces, allowing assessment of epithelial morphology, inflammatory infiltration, and corneal nerve integrity [134]. In SD, IVCM typically reveals increased dendritic cell density, which reflects active inflammation and may occur before overt clinical progression.

IVCM also reveals characteristic changes in epithelial cell morphology, including increased cell size variability, altered nuclear-to-cytoplasmic ratios, and the presence of inflammatory infiltrates. These changes correlate with disease severity and may serve as biomarkers for treatment response. The technique has also demonstrated reduced corneal nerve fiber density in SD patients, which correlates with decreased corneal sensitivity and may explain the altered pain perception seen in many patients.

Anterior segment optical coherence tomography (AS-OCT) offers high-resolution imaging of the ocular surface and can measure tear meniscus height, providing quantitative assessment of tear volume. This technique is particularly valuable for monitoring treatment response and disease progression over time. Modern AS-OCT systems can also assess conjunctival thickness and identify subclinical inflammatory changes that may not be apparent in clinical examination.

The development of tear film-specific OCT techniques has enabled dynamic assessment of tear film behavior, including measurement of tear film thinning rates and identification of areas of instability. These measurements provide objective correlates of subjective symptoms and can help guide treatment decisions.

Tear film biomarkers have emerged as promising diagnostic tools for KCS in SD, offering objective measures of inflammatory activity and glandular function. Inflammatory markers such as IL-1β, TNF-α, and matrix metalloproteinase-9 (MMP-9) are consistently elevated in SD patients and correlate with disease severity [135]. The development of point-of-care testing devices for MMP-9 has made clinical assessment of tear film inflammation more accessible [115].

Lactoferrin and lysozyme concentrations are typically reduced in SD patients, reflecting impaired lacrimal gland function [113,116]. As these proteins normally provide antimicrobial defense and maintain ocular surface integrity, their deficiency may contribute to the increased susceptibility to ocular surface infections observed in SD [118,119].

Tear osmolarity has also gained prominence as both a diagnostic and monitoring tool [136,137]. With the advent of advanced osmometers requiring only minimal sample volumes, routine clinical implementation has become feasible. Because osmolarity in SD patients fluctuates over time, protocols recommending multiple sequential measurements have been developed to better capture osmotic instability [122,123]. In addition, proteomic and metabolomic profiling of the tear film is revealing novel biomarkers with diagnostic and prognostic value, including proteins involved in inflammation, immune regulation, and tissue repair [124,125]. These approaches not only provide mechanistic insights but also identify potential therapeutic targets [128,129].

### 4.2. Functional Assessment Techniques

Dynamic assessment of tear film behavior provides insights beyond static measurements [130,131]. High-speed videokeratography can track tear film break-up patterns in real-time, revealing areas of instability and their relationship to symptom patterns [132,133]. This technique provides more detailed information than traditional TBUT testing and can identify subtle abnormalities [134].

Interferometry allows for precise measurement of the tear film lipid layer thickness and can identify patients with MGD-related evaporative dry eye [135,138]. This technique is particularly valuable in SD patients, where both aqueous deficiency and evaporative components may be present simultaneously [139].

Impression cytology enables histological examination of the conjunctival epithelium and can identify inflammatory cells, assess goblet cell density, and evaluate epithelial cell morphology [140,141]. This technique provides objective evidence of ocular surface inflammation and can help differentiate SD-related changes from other causes of KCS [142].

Questionnaire-based assessment tools such as the Ocular Surface Disease Index (OSDI) and Dry Eye Questionnaire (DEQ) provide standardized methods for symptom assessment [143,144]. These tools are particularly valuable for monitoring treatment response and can help identify patients who may benefit from additional interventions [145,146].

## 5. Current Therapeutic Approaches

A classification of therapeutic strategies for KCS in SD is provided in Table 2.

### 5.1. Artificial Tears and Lubricants

The foundation of KCS management in SD remains the replacement of deficient tear volume and improvement of tear film stability through artificial tears and lubricants. The selection of appropriate formulations requires consideration of tear film abnormalities, symptom patterns, and patient preferences. Modern artificial tears are formulated with various viscosity agents, electrolyte compositions, and osmolarity levels to address different aspects of tear film dysfunction.

The evolution of artificial tear formulations has led to increasingly sophisticated products designed to address specific tear film deficiencies. Hyaluronic acid-based formulations provide excellent mucoadhesive properties and long-lasting relief, while also promoting epithelial healing through interaction with CD44 receptors. These formulations are particularly beneficial for SD patients with significant epithelial damage.

Carboxymethylcellulose (CMC) and hydroxypropyl methylcellulose (HPMC) remain popular viscosity agents, offering good retention time and patient comfort. Newer formulations combining multiple polymers aim to provide both immediate relief and sustained protection. The addition of osmoprotectants such as L-carnitine and erythritol helps protect epithelial cells from osmotic stress.

Preservative-free formulations are strongly preferred for SD patients due to the potential for preservative toxicity in compromised ocular surfaces. Benzalkonium chloride, the most commonly used preservative in ophthalmic preparations, can cause epithelial damage, reduce goblet cell density, and exacerbate inflammatory processes. The development of multi-dose preservative-free systems using alternative preservation technologies has improved patient compliance while maintaining product sterility.

The concentration and type of electrolytes in artificial tears significantly influence their therapeutic effectiveness. Formulations with balanced electrolyte compositions that more closely mimic natural tears may provide superior comfort and healing promotion. The addition of potassium, which is typically depleted in SD patients, may help restore normal cellular function.

Hypotonic formulations may provide benefits for patients with hyperosmolar tears, though the optimal osmolarity for artificial tears remains debated. Some studies suggest that isotonic formulations may be better tolerated initially, while others indicate that slightly hypotonic solutions may help restore normal osmotic balance over time. The choice may need to be individualized based on patient response and tear film characteristics.

The frequency of artificial tear instillation must be individualized based on symptom severity, environmental factors, and tear clearance rates. Many SD patients require hourly or more frequent instillation during symptomatic periods, which can significantly impact quality of life and treatment adherence. The development of longer-lasting formulations using advanced polymer technology and lipid-containing drops has improved patient convenience and potentially enhanced therapeutic efficacy.

Gel formulations and ointments provide longer-lasting relief but may cause temporary vision blurring, making them more suitable for nighttime use. The strategic use of different viscosity products throughout the day, lighter formulations for daytime use and heavier preparations for nighttime, can optimize symptom control while minimizing visual interference.

### 5.2. Anti-Inflammatory Therapies

Topical corticosteroids may provide rapid symptom relief in keratoconjunctivitis sicca, particularly during acute inflammatory flares. However, their use should be limited to short-term courses given the well-recognized risks of long-term therapy, including glaucoma, cataract formation, and ocular surface complications.

Among steroid-sparing options, topical cyclosporine and lifitegrast have emerged as the mainstays of anti-inflammatory therapy. Cyclosporine, a calcineurin inhibitor, reduces T-lymphocyte activation and inflammatory cytokine production, while lifitegrast, an antagonist of lymphocyte function–associated antigen-1 (LFA-1), interrupts T-cell adhesion and migration. Both agents directly address the underlying autoimmune inflammation, albeit through distinct mechanisms. Clinical trials have demonstrated that cyclosporine 0.05% emulsion, administered twice daily, improves tear production and symptoms, although the onset of action may require several weeks. Lifitegrast, in contrast, has shown improvement in both signs and symptoms within the first few weeks of treatment [147]. This calcineurin inhibitor reduces T-lymphocyte activation and inflammatory cytokine production, addressing the underlying autoimmune process. Clinical trials have demonstrated significant improvements in tear production and symptom relief with twice-daily cyclosporine 0.05% emulsion. The onset of action is typically gradual, requiring several weeks to months for optimal benefit.

Topical lifitegrast, a lymphocyte function-associated antigen-1 (LFA-1) antagonist, represents a novel approach to anti-inflammatory therapy [148,149]. By blocking T-cell activation and migration, lifitegrast interrupts the inflammatory cascade at a different point than cyclosporine. Clinical studies have shown improvements in both signs and symptoms of KCS, with benefits often apparent within the first few weeks of treatment.

The development of newer anti-inflammatory agents continues to expand therapeutic options. Topical corticosteroids with improved safety profiles, such as loteprednol etabonate, may provide anti-inflammatory benefits with reduced systemic absorption and fewer side effects.

### 5.3. Secretagogues and Tear Stimulants

Oral pilocarpine, a muscarinic agonist, stimulates residual lacrimal gland function and can increase tear production in SD patients with remaining functional glandular tissue [150,151]. The drug is most effective in patients with mild to moderate glandular dysfunction and may provide additional benefits for salivary gland function. Side effects include sweating, nausea, and urinary frequency, which limit its use in some patients.

Cevimeline, another muscarinic agonist, offers similar benefits to pilocarpine but with potentially improved tolerability [150]. The drug has shown efficacy in increasing tear production and reducing symptoms in SD patients, though individual responses vary considerably.

Topical secretagogues are under investigation as potential alternatives to systemic therapy. These agents could theoretically provide localized stimulation of tear production without systemic side effects, though clinical development remains in early stages.

Diquafosol, a P2Y2 receptor agonist, stimulates mucin and fluid secretion from conjunctival epithelial cells and goblet cells. While not available in all markets, this agent has shown promise in clinical trials for improving tear film stability and reducing symptoms.

### 5.4. Procedural Interventions

Punctal occlusion offers a mechanical strategy for tear conservation and is particularly useful in patients with severe aqueous deficiency [43,152]. Temporary collagen plugs are often placed initially to evaluate benefit before considering permanent silicone plugs. The procedure is generally safe and well tolerated, although potential complications include plug extrusion, infection, and epiphora.

Adjunctive mechanical techniques such as lacrimal gland massage and expression may provide short-term symptom relief by promoting glandular secretion, but their long-term efficacy remains limited.

Intense pulsed light (IPL) therapy has emerged as a novel approach for meibomian gland dysfunction–associated evaporative dry eye in SD. By targeting abnormal periocular vessels, IPL may improve gland function, though additional studies are required to establish standardized treatment protocols.

For patients with severe, refractory keratoconjunctivitis sicca, scleral contact lenses provide a protective fluid reservoir over the cornea, improving ocular surface hydration and reducing symptom burden [153]. Although these devices require specialized fitting and close follow-up, they can substantially enhance quality of life.

## 6. Emerging Therapeutic Strategies

### 6.1. Biologic Therapies and Targeted Interventions

The development of biologic therapies represents a paradigm shift in KCS treatment, offering the potential to target specific inflammatory pathways involved in SD pathogenesis. These treatments address the underlying autoimmune mechanisms rather than merely providing symptomatic relief, potentially offering disease-modifying effects that could slow or halt disease progression.

Rituximab, a monoclonal antibody targeting CD20+ B cells, has shown promise in multiple studies of SD patients, with improvements in both systemic and ocular manifestations [154,155]. Clinical trials have demonstrated improvements in tear production, ocular surface staining, and patient-reported symptoms following rituximab treatment. Yet these benefits are often temporary, necessitating repeated courses. Moreover, the use of rituximab is constrained by its high cost, limited access in many healthcare systems, and the increased risk of infections associated with prolonged B-cell.

The mechanism of action of rituximab in SD extends beyond simple B-cell depletion. The treatment appears to reset the immune system, reducing the production of autoantibodies and pro-inflammatory cytokines. Some patients experience prolonged remissions following treatment, suggesting that rituximab may help restore immune tolerance in certain individuals.

TNF-α inhibitors, including infliximab, etanercept, and adalimumab, have been investigated for SD treatment with mixed results [156,157]. While some studies have shown improvements in systemic symptoms, ocular benefits have been less consistent. The heterogeneity of SD patients and varying degrees of TNF-α involvement may explain these disparate results. Some patients may benefit significantly from TNF-α inhibition, while others show minimal response.

The use of TNF-α inhibitors in SD requires careful patient selection and monitoring. These agents carry risks of serious infections and may exacerbate certain autoimmune conditions. The decision to use TNF-α inhibitors should be based on disease severity, failure of conventional therapy, and careful risk-benefit assessment.

Abatacept, a selective co-stimulation modulator that blocks T-cell activation, has shown promising results in early clinical trials for SD [158]. By interrupting the interaction between CD80/CD86 on antigen-presenting cells and CD28 on T cells, abatacept may help reduce the autoimmune response responsible for glandular destruction. The drug has demonstrated improvements in both glandular function and systemic symptoms in some patients.

The advantage of abatacept lies in its mechanism of action, which targets the fundamental T-cell activation process central to SD pathogenesis. Unlike treatments that target downstream effector molecules, abatacept may address the root cause of autoimmune activation. However, like other biologic therapies, it requires careful monitoring for adverse effects and may not be suitable for all patients.

Interferon-α therapy has demonstrated efficacy in treating systemic manifestations of SD and may have ocular benefits [159]. The treatment appears to modulate immune function and may help restore glandular function in some patients. However, the significant side effect profile of interferon therapy, including flu-like symptoms, depression, and autoimmune complications, limits its use to carefully selected patients with severe disease.

Newer biologic agents targeting other inflammatory pathways are under investigation. Interleukin-6 inhibitors, such as tocilizumab, have shown promise in preliminary studies. IL-6 plays a crucial role in B-cell activation and autoantibody production, making it an attractive therapeutic target. Other potential targets include the BAFF (B-cell activating factor) pathway, which is important for B-cell survival and activation.

The development of biomarkers to predict responses to biologic therapy is an active area of research. Factors such as autoantibody profiles, cytokine levels, and genetic markers may help identify patients most likely to benefit from specific treatments. This personalized approach could improve treatment outcomes while reducing unnecessary exposure to expensive and potentially harmful therapies.

### 6.2. Regenerative Medicine Approaches

Stem cell therapy represents a promising frontier in KCS treatment, with potential applications for both lacrimal gland regeneration and ocular surface repair. Mesenchymal stem cells (MSCs) derived from various sources, including bone marrow, adipose tissue, and umbilical cord, have shown anti-inflammatory and regenerative properties in preclinical studies. Recent clinical trials have demonstrated the potential of allogeneic stem cell therapy in treating immune-mediated keratoconjunctivitis sicca, with significant improvements in Schirmer test scores observed in canine models of autoimmune dry eye disease [160,161]. The mechanisms underlying stem cell therapeutic effects include paracrine secretion of growth factors, immunomodulatory cytokines, and extracellular vesicles that promote tissue regeneration and reduce local inflammation [162,163].

Advanced tissue engineering approaches are being developed for lacrimal gland reconstruction, including bioengineered organoids that recapitulate the complex three-dimensional architecture of native glands [164,165]. These organoid systems have shown promise in preclinical studies for restoring secretory function and may eventually provide alternatives to conventional medical therapy for patients with severe glandular destruction [166,167].

Autologous serum eye drops have gained popularity as a treatment for severe KCS, providing growth factors, vitamins, and other biological components that may promote epithelial healing [33]. While evidence for efficacy is limited, many patients report subjective improvements, and the treatment is generally well-tolerated.

Platelet-rich plasma (PRP) eye drops represent an evolution of serum therapy, providing concentrated growth factors and cytokines that may enhance healing. Early clinical studies suggest potential benefits, though standardization of preparation methods and treatment protocols remains challenging. Recent advances in PRP preparation techniques have improved standardization, with studies showing that PRP eye drops containing optimal concentrations of platelet-derived growth factor (PDGF) and transforming growth factor-β (TGF-β) provide superior healing outcomes compared to conventional artificial tears [168,169]. The development of autologous plasma-rich in growth factors (PRGF) represents a further refinement of this approach, offering enhanced bioavailability and sustained release of therapeutic factors [170,171].

Tissue engineering approaches for lacrimal gland reconstruction are in early developmental stages. The complex structure and function of the lacrimal gland present significant challenges, but advances in 3D bioprinting and organoid technology may eventually enable gland replacement or augmentation.

### 6.3. Gene Therapy and Molecular Interventions

Gene therapy approaches for KCS are being investigated, with potential applications including enhancement of tear production, reduction in inflammation, and promotion of epithelial healing [61,172]. Viral vectors could theoretically deliver therapeutic genes to lacrimal glands or ocular surface tissues, providing sustained therapeutic effects [173,174]. Recent advances in adeno-associated virus (AAV) vector technology have improved targeting specificity and reduced immunogenicity, making clinical translation more feasible [175,176].

Epigenetic modulation represents an emerging therapeutic strategy, with studies identifying specific DNA methylation patterns and histone modifications associated with SS pathogenesis [177,178]. Targeting these epigenetic changes through selective inhibitors may offer new approaches to restoring normal gene expression patterns in affected tissues [168].

Antisense oligonucleotides and small interfering RNAs (siRNAs) offer precise methods for modulating gene expression in target tissues. These approaches could potentially reduce inflammatory mediator production or enhance protective factor expression in the ocular surface [179,180,181].

Nanotechnology applications in KCS treatment include drug delivery systems that can provide sustained release of therapeutic agents and targeted delivery to specific ocular surface tissues. Nanoparticles, liposomes, and other delivery systems may enhance therapeutic efficacy while reducing systemic exposure. Recent developments in nanocarrier systems include thermosensitive hydrogels that provide prolonged drug residence time on the ocular surface, and targeted nanoparticles that can selectively deliver anti-inflammatory agents to activated immune cells [182,183]. These advanced delivery systems show promise for improving treatment outcomes while minimizing side effects [182,184].

## 7. Future Directions and Research Priorities

### 7.1. Personalized Medicine Approaches

The heterogeneity of SD patients and their varying responses to treatment highlight the need for personalized medicine approaches that consider individual patient characteristics, disease phenotypes, and molecular profiles [13,185]. This approach represents a fundamental shift from the traditional “one-size-fits-all” treatment paradigm to tailored interventions based on specific patient needs and disease mechanisms [186,187,188].

Biomarker-guided therapy selection is becoming increasingly important in SD management, with recent advances in tear proteomics and metabolomics enabling identification of patient-specific inflammatory profiles for targeted treatment selection [189,190,191]. Patients with high levels of specific inflammatory markers may benefit more from anti-inflammatory therapies, while those with evidence of glandular dysfunction may require secretagogues or regenerative approaches [192,193]. The development of point-of-care testing devices for key biomarkers is making personalized treatment selection more feasible in clinical practice [194,195].

Genetic profiling may reveal susceptibility factors and therapeutic targets specific to individual patients. Polymorphisms in genes involved in drug metabolism, such as cytochrome P450 enzymes, could influence optimal dosing of medications like cyclosporine [196,197]. Additionally, genetic variants in inflammatory pathways may predict response to specific anti-inflammatory treatments [198].

Endotypes are one of the most novel concepts emerging in SD research is the recognition of disease *endotypes*—biologically distinct subgroups characterized by different underlying mechanisms [199,200]. This framework moves beyond clinical phenotyping toward mechanism-based stratification. Patients with predominantly inflammatory endotypes may respond preferentially to immunosuppressive therapies, while those with neurogenic or glandular dysfunction endotypes may require alternative interventions [135,201]. Identification of endotype-specific biomarkers could revolutionize therapeutic algorithms, and they represent a key step toward precision medicine in SD [187].

Pharmacogenomics applications in SD are still in early stages but hold promise for optimizing treatment selection and dosing [202]. Genetic variants affecting drug absorption, metabolism, and target receptor expression could influence treatment outcomes [203]. The integration of pharmacogenomic data into clinical decision-making tools could improve treatment efficacy while reducing adverse effects.

Tear film proteomic and metabolomic analyses are providing new insights into disease mechanisms and potential therapeutic targets [204]. These approaches may eventually enable the development of personalized treatment protocols based on individual tear film profiles. The identification of specific metabolic pathways that are dysregulated in individual patients could guide targeted interventions.

**Patient-centered outcomes:** The refinement of patient-reported outcome measures is essential to capture the heterogeneous impact of SD and treatment response. Personalized symptom profiles, aligned with individual quality-of-life priorities, may further guide therapy and monitoring [205,206].

**Systems medicine and AI:** Integration of multi-omic data (clinical, genetic, biochemical, and imaging) using artificial intelligence and machine learning is being developed to create predictive models of treatment response and long-term outcomes [207,208]. Recent applications in ophthalmology illustrate this potential: for example, deep learning algorithms applied to in vivo confocal microscopy and optical coherence tomography (OCT) have been shown to accurately classify dry eye severity and detect subtle corneal nerve alterations before clinical symptoms emerge. For instance, a recent deep learning model applied to anterior segment optical coherence tomography (AS-OCT) images demonstrated robust diagnostic performance for dry eye disease, achieving 85% accuracy, 86% sensitivity, and 82% specificity when compared to traditional clinical tests—suggesting that AI can provide fast, objective, and reproducible assessments of tear film abnormalities. Such tools could assist clinicians in delivering more precise, individualized care in SD. Emerging AI techniques are also being applied to in vivo confocal microscopy (IVCM). A novel machine learning algorithm now enables unbiased quantification of corneal nerve features—such as nerve density and dendritic cell infiltration—offering objective biomarkers that correlate with ocular surface inflammation.

Collectively, these advances highlight a paradigm shift toward precision medicine in SD. By integrating biomarkers, genetics, and computational tools, future treatment strategies may be tailored to individual disease mechanisms, with the ultimate goal of improving outcomes while reducing treatment burden.

### 7.2. Combination Therapy Strategies

The complex pathophysiology of KCS in SD suggests that combination therapies targeting multiple pathways may be more effective than single-agent approaches. Combinations of anti-inflammatory agents, tear replacements, and procedural interventions may provide synergistic benefits. Recent studies have shown that combining topical cyclosporine with punctal occlusion provides superior outcomes compared to either treatment alone.

Network medicine approaches are revealing unexpected therapeutic targets through analysis of protein–protein interaction networks and pathway crosstalk. The integration of traditional Chinese medicine with Western pharmacological approaches has shown promise in clinical trials, suggesting that combination strategies may benefit from both Eastern and Western therapeutic traditions.

Sequential therapy protocols that adapt treatment based on patient response and disease progression may optimize outcomes while minimizing side effects. The development of treatment algorithms incorporating objective measures of treatment response could guide therapy selection and adjustment.

## 8. Conclusions

Keratoconjunctivitis sicca (KCS) in Sjögren disease represents a complex autoimmune condition that extends far beyond simple tear deficiency. Its multifactorial pathophysiology involves immune, inflammatory, hormonal, and neurogenic mechanisms, making effective management particularly challenging.

Recent advances in diagnostics, including high-resolution imaging and biomarker discovery, are improving early detection and enabling more objective disease monitoring. Therapeutic strategies have evolved from symptomatic tear replacement to targeted anti-inflammatory agents, procedural interventions, and emerging biologic and regenerative therapies, with the potential to modify disease progression rather than simply alleviate symptoms. If validated in larger trials, these innovations could ultimately reshape international treatment guidelines.

Among the most novel concepts highlighted in this review is the recognition of disease endotypes—biologically distinct subgroups of Sjögren disease defined by their underlying mechanisms rather than clinical manifestations. This framework represents a paradigm shift toward mechanism-based classification, offering the potential to refine therapeutic selection, accelerate personalized medicine, and fundamentally alter future treatment algorithms.

Looking ahead, the integration of endotype-specific biomarkers, genetic and pharmacogenomic data, and computational approaches such as artificial intelligence will pave the way toward precision medicine in Sjögren disease. Multidisciplinary collaboration between ophthalmologists, rheumatologists, and other specialists, combined with patient-centered care models, will be essential for translating these scientific advances into improved outcomes and quality of life for patients.

## Figures and Tables

**Figure 1 ijms-26-08824-f001:**
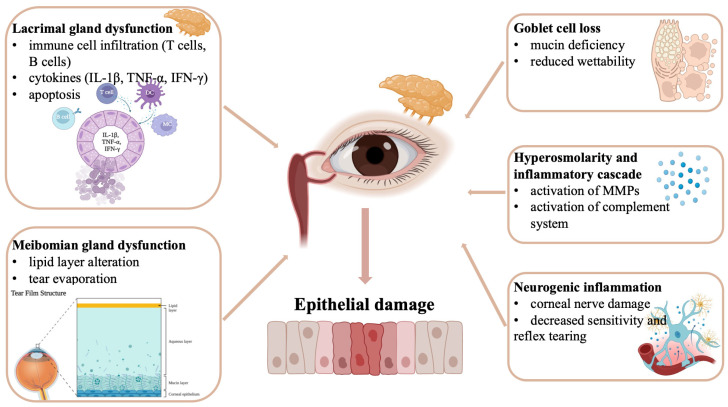
Pathophysiology of keratoconjunctivitis sicca in Sjögren disease. Keratoconjunctivitis sicca arises from lacrimal gland dysfunction, meibomian gland abnormalities, goblet cell loss, and tear film hyperosmolarity. These defects, amplified by immune-mediated inflammation and corneal nerve damage, culminate in chronic ocular surface inflammation and epithelial injury.

**Table 1 ijms-26-08824-t001:** Diagnostic tools for keratoconjunctivitis sicca in Sjögren Disease.

Test/Tool	Type	Target Parameter	Advantages	Limitations
Schirmer test	Functional	Tear volume	Simple, widely available	Poor specificity, variable reproducibility
TBUT	Functional	Tear film stability	Non-invasive, fast	Operator-dependent
Vital dye staining	Structural	Epithelial integrity	Detects surface damage	Variable correlation with symptoms
Tear osmolarity	Biomarker	Tear composition (osmolarity)	Quantifiable, point-of-care devices	Fluctuates; sensitive to environment
MMP-9 assay (InflammaDry)	Biomarker	Ocular surface inflammation	Fast, POC available	Binary output, not quantifiable
Meibography	Imaging	Meibomian gland morphology	Structural assessment	Limited availability
In vivo confocal microscopy	Imaging	Inflammatory cells, nerve density	High-resolution, detailed	Requires expertise, limited access
Anterior segment OCT	Imaging	Tear meniscus, conjunctiva	Objective, reproducible	Still not routine in many clinics
OSDI/DEQ	Symptom-based	Patient-reported symptom severity	Easy to administer	Subjective, no correlation with signs

Abbreviations: AS-OCT: Anterior Segment Optical Coherence Tomography; DEQ: Dry Eye Questionnaire; IVCM: In Vivo Confocal Microscopy; MMP-9: Matrix Metalloproteinase-9; OSDI: Ocular Surface Disease Index; POC: Point-of-Care; TBUT: Tear Break-Up Time.

**Table 2 ijms-26-08824-t002:** Therapeutic approaches for keratoconjunctivitis sicca in Sjögren Disease.

Treatment Class	Examples	Mechanism of Action	Indication/Utility	Limitations/Notes
Artificial tears	Hyaluronic acid, CMC, lipid-based drops	Tear replacement, lubrication	First-line for all severities	Temporary relief, frequent use needed
Topical corticosteroids	Loteprednol, fluorometholone	Inhibit inflammation	Short-term flare control	Risk of IOP rise, cataract with prolonged use
Topical immunomodulators	Cyclosporine, lifitegrast	T-cell inhibition (calcineurin/LFA-1 pathways)	Chronic inflammation, maintenance therapy	Delayed onset; stinging on instillation
Oral secretagogues	Pilocarpine, cevimeline	Muscarinic receptor agonists	Residual gland function	Cholinergic side effects
Punctal occlusion	Silicone or collagen plugs	Reduces tear drainage	Moderate-to-severe aqueous deficiency	Epiphora, plug extrusion
Biologic therapies	Rituximab, abatacept, tocilizumab	Target B/T cells, cytokines (CD20, CD80/86, IL-6)	Refractory systemic and ocular disease	Off-label; systemic risks; cost
Regenerative therapy	Autologous serum, PRP, stem cells	Growth factors, epithelial healing	Severe epithelial damage, neurotrophic KCS	Access, standardization challenges
Procedural	IPL, scleral lenses	MGD treatment, tear reservoir creation	Severe or refractory cases	Requires expertise; cost

Abbreviations: CMC: Carboxymethylcellulose; IOP: Intraocular Pressure; LFA-1: Lymphocyte Function-Associated Antigen-1; PRP: Platelet-Rich Plasma; NSAID: Nonsteroidal Anti-Inflammatory Drug; MGD: Meibomian Gland Dysfunction; KCS: Keratoconjunctivitis Sicca; SD: Sjögren Disease.

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
