# Peer review of "Keratoconjunctivitis Sicca in Sjögren Disease: Diagnostic Challenges and Therapeutic Advances"

_ijms, 2025, doi:10.3390/ijms26188824_

Round 1
Reviewer 1 Report
Comments and Suggestions for Authors
Comments:
1.most of the paragraphs have no more than 2 references. I wonder why? Section6.2-whole section has no references. Future directions and research priorities- the whole section, one whole page has no references. Even if this section is author’s perspective, it still needs to be backed up by available literature for the topic. Please cite multiple references. This is a review article and personally I think 76 references are too low. Try to cite multiple references.
2.Latest reference cited in the manuscript is from 2019. Please make it up-to-date.
- In section 2.1-glandular dysfunction has not been discussed. Please provide some details of glandular dysfunction or change the title
- section2.3- how hyperosmolarity activate inflammatory pathways? Please include more insight rather than just mentioning it.write something about how osmolarity can affect cellular functions of epithelial cells or goblet cells. There must be some event leading to proinflammatory cytokine secretion. Otherwise, the whole manuscript looks very repeatitive.
- section 4.1- imaging technologies and biomarkers- line 249- advanced meibography calculate gland dropout scores and tortuosity indices. Explain and elaborate how these indices are important. Define tortuosity here-is it for gland or blood vessels. How high index is different than low index etc. Again, make it more informative and insightful.
Author Response
Dear Reviewer,
We thank you for your comments and suggestions. We acknowledge your remarks . The list of references have been extensively updated and suggestions implemented in the text .
KR
Shan
Reviewer 2 Report
Comments and Suggestions for Authors
The manuscript “Keratoconjunctivitis Sicca in Sjögren's Disease: Diagnostic Challenges and Therapeutic Advances” provides a comprehensive and detailed overview of the pathophysiology, diagnosis, and therapeutic options for KCS in the context of Sjögren’s disease. The work is timely, well-structured, and covers both established and emerging treatment modalities. The review demonstrates depth of knowledge and cites a wide range of recent literature.
However, there are areas where the manuscript could be improved for clarity, conciseness, and flow. Some sections are overly dense with technical details without enough synthesis. Figures and tables are useful but could be better integrated into the narrative. The introduction and conclusion should emphasize novelty and the significance of this review compared to existing literature. A few sections require clearer transitions, reduction of redundancy, and tightening of language.
- Abstract
- The abstract is well-written but quite dense. Consider simplifying technical terms for a broader audience.
- Suggest shortening sentence at line 17–19 (“Aberrant activation of both innate and adaptive immune systems…”) to improve readability.
- Introduction
- Line 33: “stills remains shrouded” → grammatical correction: “still remains unclear.”
- Line 36–39: Consider splitting the long sentence for clarity (epidemiology + symptoms separately).
- Line 48–50: The economic burden section is strong; adding a quantified cost estimate (if available from literature) would strengthen it.
- Pathophysiology
- Lines 75–81: The description of molecular mimicry and autoantigens could be simplified. Suggest condensing to highlight only key autoantigens (Ro52, Ro60, La).
- Lines 92–100: Long list of cytokines—consider grouping (“pro-inflammatory cytokines such as TNF-α, IL-1β, IFN-γ”).
- Figure 1 (Line 152–161): The legend is long; could be streamlined into shorter sentences.
- Diagnostic Challenges
- Lines 170–175: “discordance between symptoms and signs” is a strong point. Suggest adding one recent reference (post-2020) to support this.
- Lines 188–194: The psychological burden is important—could benefit from its own short paragraph emphasizing quality-of-life studies.
- Advanced Diagnostics
- Lines 256–268: In vivo confocal microscopy section is very detailed—suggest condensing to 3–4 sentences.
- Table 1 (Line 317): Excellent summary, but could be improved visually with clearer separation between “advantages” and “limitations.”
- Therapeutics
- Lines 369–373: Topical corticosteroids—add cautionary note about short-term vs long-term use (risk of glaucoma, cataracts).
- Lines 377–384: Cyclosporine and lifitegrast—combine into one paragraph to avoid repetition of “anti-inflammatory mechanism.”
- Lines 405–421: Procedural interventions—this section could be reorganized with subheadings (e.g., “punctal occlusion,” “scleral lenses,” “light therapy”).
- Emerging Therapies
- Lines 429–435: Rituximab—add a sentence on limitations (cost, access, infection risk).
- Lines 468–477: Personalized biomarkers—consider linking to the “Future directions” section for flow.
- Lines 478–491: Regenerative medicine—add note on regulatory/ethical considerations for stem cell therapy.
- Future Directions
- Line 520–536: The “endotypes” concept is important—highlight this as one of the novel contributions of the review.
- Line 551–553: Machine learning mention could benefit from a brief example of recent application in ophthalmology.
- Conclusion):
- Lines 567–575: Too many repeated phrases like “complex,” “holistic,” “comprehensive.” Suggest tightening.
- Lines 590–599: Biologics and regenerative therapy discussion is strong. Suggest adding one short sentence on how this could change treatment guidelines in the future.
- Minor Corrections:
- Line 33: “stills remains” → “still remains.”
- Line 41: “kCS” should be “KCS” (typo).
- Ensure uniform formatting of abbreviations (e.g., always define TNF-α at first mention, consistent use of “IL-1β” vs “interleukin-1β”).
Comments on the Quality of English Language
The English could be improved to more clearly express the research.
Author Response
Dear Reviewer ,
We thank you for your useful comments and remarks that have been thoroughly implemented in the revised text.
Kind regards
Shan
Round 2
Reviewer 2 Report
Comments and Suggestions for Authors
Manuscript ready For publicTion